# Impact of Soil Conservation and Eucalyptus on Hydrology and Soil Loss in the Ethiopian Highlands

**Demesew A. Mhiret** [1,2], **Dessalegn C. Dagnew** [2,3], **Tilashwork C. Alemie** [4], **Christian D. Guzman** [5], **Seifu A. Tilahun** [1], **Benjamin F. Zaitchik** [6] and **Tammo S. Steenhuis** [1,7,*]

[1]  Faculty of Civil and Water Resources Engineering, Bahir Dar Institute of Technology, Bahir Dar University, Bahir Dar, Ethiopia; demisalmaw@gmail.com (D.A.M.); satadm86@gmail.com (S.A.T.)
[2]  Blue Nile Water Institute, Bahir Dar University, Bahir Dar, Ethiopia; cdessalegn@yahoo.com
[3]  Institute of Disaster Risk Management and Food Security Studies, Bahir Dar University, Bahir Dar, Ethiopia
[4]  Amhara Regional Agricultural Research Institute, Bahir Dar, Ethiopia; tilashworkchanie@gmail.com
[5]  Department of Civil and Environmental Engineering, University of Massachusetts, Amherst, MA 01003, USA; cdguzman@umass.edu
[6]  Department of Earth and Planetary Sciences, Johns Hopkins University, Baltimore, MD 21218, USA; zaitchik@jhu.edu
[7]  Department of Biological and Environmental Engineering, Cornell University, Ithaca, NY 14853, USA
[*] Correspondence: tss1@cornell.edu; Tel.: +1-607-255-2489

**Abstract:** The Ethiopian highlands suffer from severe land degradation, including erosion. In response, the Ethiopian government has implemented soil and water conservation practices (SWCPs). At the same time, due to its economic value, the acreage of eucalyptus has expanded, with croplands and pastures converted to eucalyptus plantations. The impact of these changes on soil loss has not been investigated experimentally. The objective of this study, therefore, is to examine the impacts of these changes on stream discharge and sediment load in a sub-humid watershed. The study covers a nine-year period that included installation of SWCPs, a three-fold increase from 1.5 ha in 2010 to 5 ha in 2018 in eucalyptus, and the upgrading of an unpaved to the paved road. Precipitation, runoff, and sediment concentration were monitored by installing weirs at the outlets of the main and four nested watersheds. A total of 867 storm events were collected in the nine years. Runoff and sediment concentration decreased by more than half in nine years. In the main watershed W5, we estimated that evapotranspiration by eucalyptus during the dry phase (November to May) increased approximately from 30 mm a$^{-1}$ in 2010 to 100 mm a$^{-1}$ in 2018. In watershed W3 it increased from 2 mm a$^{-1}$ to 400 mm a$^{-1}$, requiring more rainfall before saturation excess runoff began in the rain phase. The reduction in runoff led to a decreased sediment load from 70 Mg ha$^{-1}$ a$^{-1}$ in 2010 to 2.8 Mg ha$^{-1}$ a$^{-1}$ in 2018, though the reduction in discharge may have negative impacts on ecology and downstream water resources. SWCPs became sediment-filled and minimally effective by 2018. This indicates that these techniques are either inappropriate for this sub-humid watershed or require improved design and maintenance.

**Keywords:** Ethiopian highlands; eucalyptus; gully; soil loss; soil and water conservation practices

## 1. Introduction

Land degradation and associated soil loss is a major global ecological problem [1–4]. The Ethiopian highlands are especially degraded [4–6]. Land degradation is accelerated by anthropogenic factors, such as population growth, cultivation of steep slopes, clearing of vegetation, overgrazing, and increased soil erosion [5,7–9].

In response to the severe drought in the 1970s, the Ethiopian government started to implement soil and water conservation practices (SWCPs) to reverse the trend in soil degradation. In 2012, the government expanded its conservation efforts, requiring rural farmers to volunteer their labor in January and February each year to install centrally planned SWCPs. The effectiveness of these practices is being debated [10]. In the semi-arid highlands, SWCPs perform well in conserving moisture and generally increase crop yields [3]. In the sub-humid and humid highland regions, with rainfall in excess of potential evaporation during the rainy phase, conserving moisture is not a priority. Instead, preventing saturation of the root zone is a priority. Consequently, the purpose of SWCPs is to safely remove the excess water [11,12]. Practices that increase infiltration, such as Fanya- Juu ("throw uphill") bunds, infiltration furrows, and stone bunds have been shown to be effective in decreasing soil losses in the first five years after implementation [9,13] except in watersheds with gullies downstream [14,15]. Gullies have been identified as a critical factor in soil loss from catchments in the sub-humid Ethiopian highlands [12,14,16,17].

Gully erosion is largely a consequence of forests being replaced by agricultural lands that are cultivated year after year [12]. The continuous cultivation following deforestation decreases organic matter content, that causes soil degradation and hardpan formation [16]. The hard pan reduces deep percolation rates and increases the interflow and surface runoff [12,16,18]. This in turn results in soil saturation in valley bottoms, reducing the cohesive soil strength and enhancing gully formation.

One approach that has been applied to understand relationships between runoff (Q) and suspended sediment concentration (SSC) in the presence of SWCPs is the analysis of hysteresis loops [19–21]. Analysis of the Q–SSC relationship indicated five hysteresis loop patterns that can be reduced to three categories [20]: clockwise, counter-clockwise, and mixed loops. Clockwise loops occur when the concentration on the rising limb is greater than the concentration on the recession limb at equal discharge and are characterized by a sediment peak before the discharge peak. It indicates that sediment becomes more limited during the storm [22]. Mixed loops are those where either the concentration is relatively constant throughout the runoff event or small sediment peaks occur before and after the discharge peak. Mixed loops occur when sediment is equally available during the runoff event [23–25]. Counter-clockwise loops have a greater concentration during the recession than during the rising limb and indicate that a new sediment source is accessed during the recession. This can be related to gully banks failing during the storm [23]. The fact that gully bank erosion has a specific, process-relevant signal in hysteresis loops suggests that analysis of these loops may be useful in assessing the relative roles of upland soil erosion and gully erosion in the presence of SWCPs and other land-use change. We note that in some contexts, the travel time from a distant sediment source has been invoked as an explanatory variable for the type of hysteresis loop [21,26,27], but this explanation does not apply in small watersheds with sampling intervals less than the time of concentration. In the Birr watershed of the upper Blue Nile basin, gully rehabilitation through planting vegetation (e.g., *Sesbania sesban*) was proven effective [28].

The objective of this study is to investigate the impact of the implementation of SWCPs and the expansion of eucalyptus acreage on discharge and soil loss patterns. The study was carried out in the 95-ha Debre Mawi watershed in the sub-humid Ethiopian highlands. In the main watershed, four nested watersheds were defined. Discharge and sediment loss were monitored at each watershed by installing a weir at the outlets. Samples were collected at 10-min time step during runoff events. The hysteresis patterns of suspended sediment concentrations and discharge were identified and analyzed in each watershed. Discharge and sediment concentration and loads were compared and evaluated.

## 2. Materials and Methods

### 2.1. Description of the Study Area

The Debre Mawi watershed (Figure 1) is located in the northern Ethiopian highlands, Amhara Regional State, 30 km from Bahir Dar on the road to Adet. The outlet is located at 37°22′11″ E and

11°18′17″ N. The elevation ranges from 2195 m near the outlet to 2308 m in the southeast. Slopes vary from 1% to 30%. The climate is categorized as sub-humid with a mean annual rainfall of 1240 mm a$^{-1}$ [29]. Seventy percent of the rainfall falls from June to September. The mean daily temperature is 20 °C. Four smaller watersheds were nested in the main 95 ha watershed (Figure 1). These watersheds were W1 (8.8 ha), W2 (11.0 ha), W3 (6.4 ha), and W4 (10.4 ha). The entire watershed was named W5. The weirs at the outlets were given the same number as the watershed. So W1 was monitored by weir 1 and so on.

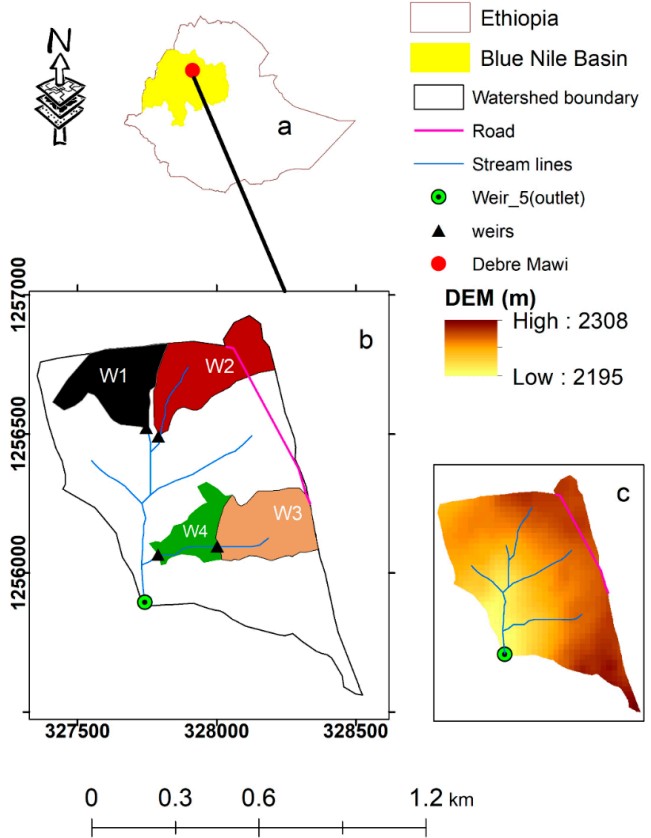

**Figure 1.** Location of Debre Mawi watershed and map of its four nested watersheds: (**a**) The location of the Debre Mawi watershed in the Upper Blue Nile Basin. (**b**) The Debre Mawi watershed consisting of the main watershed W5 together with the nested watersheds W1, W2, W3, and W4. The coordinates are in UTM. (**c**) The digital elevation map of the Debre Mawi watershed.

The lower part of the soil profile consists of shallow, highly weathered and fractured basalt overlain by dark-brown clay and light-brown, wet, sticky-clay. Intrusive dikes perpendicular to the flow direction blocks subsurface flow and results in the emergence of springs during the wet season [17,30,31]. The major types of soil in the watershed are Nitisols, Vertisols, and Vertic Nitisols [32]. Nitisols are fertile forest soils with high base saturation (>35%) and red, clay-loam soils covering the upper part of the watershed. These are very deep, well-drained, permeable soils and are suited for cereal cultivation. Vertisols, characterized by expanding montmorillonite clay, are found at the bottom part of the watershed [17]. They form shrinkage cracks on drying and swell during the rainy season. The mid-slope of the watershed is dominated by reddish-brown Vertic Nitisols with good permeability and high moisture retention capacity. These soils are particularly well suited for '*teff*' (*Eragrostis abyssinica*) production [31].

During the experimental period, farmers planted eucalyptus trees, and the acreage of trees increased from 1.5 ha to 5 ha (Figure 2). By 2018, 74% of the watershed was cropped, 15% grass and 5% eucalyptus trees, and 6% sparse vegetation. The expansion of eucalyptus was mainly on the cropland

in Watershed W3 and on the erosive-prone land near the outlet of the main watershed. In addition, the small trees in 2010 were full-grown by 2018.

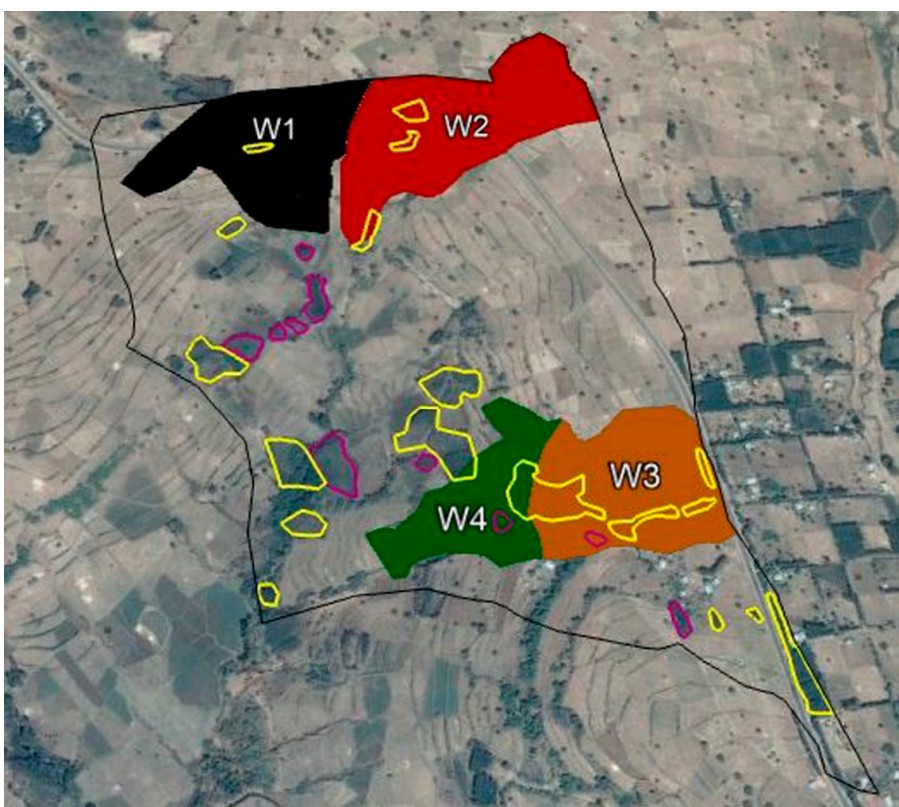

**Figure 2.** Eucalyptus tree expansion in the Debre Mawi Watershed (Google earth). Black is the boundary of the main watershed; purple indicates the eucalyptus area in 2010, and yellow indicates additional eucalyptus by 2018. W1, W2, W3, and W4 are the nested watersheds in the main watershed W5.

The grasslands are found in the valley bottomlands, which are too wet for cropping during the rain phase. Shrubs are found on the stony, steep, and shallow soils on the hillslopes. Crops grown are *teff*, maize (*Zea mays*), finger millet (*Eleusine coracana*) barley (*Hordeum vulgare*), and wheat (*Triticum aestivum*) [10,14,16]. The small nested watersheds W2 and W4 had a land use distribution that is comparable with the main watershed. Watershed W1 that had the largest portion of grassland and Watershed W3 had the largest eucalyptus acreage (Table 1).

**Table 1.** The land cover acreage (ha) in Debre Mawi and nested sub-watersheds in 2010 and 2018. Grassland did not change, so only one value is given.

| Water-Shed | Cultivated | | Grass Land | Shrub | | Eucalyptus | | Total | Existence of Gully |
|---|---|---|---|---|---|---|---|---|---|
| | 2010 | 2018 | | 2010 | 2018 | 2010 | 2018 | | |
| 1 | 3.0 | 2.8 | 5.2 | 0.6 | 0.6 | 0 | 0.2 | 8.8 | No gully |
| 2 | 8.0 | 7.3 | 2.6 | 0.4 | 0.7 | 0 | 0.7 | 11 | Upland gully formed in 2017 |
| 3 | 5.1 | 4.1 | 0.6 | 0.7 | 0.7 | 0.1 | 1.1 | 6.5 | No gully |
| 4 | 8.0 | 7.6 | 0.9 | 1.5 | 1.5 | 0 | 0.4 | 10.4 | Gully became stable |
| 5 | 69 | 67.5 | 14 | 10.5 | 8.5 | 1.5 | 5 | 95 | 15 small gullies; 1 gully of 1500 m$^2$. |

Five-meter-deep and twenty-meter-wide gullies can be seen in the valley bottomlands (Figure 3a). Formation of the gullies in the Debre Mawi watershed started in the 1980s following the removal

of indigenous forests, which in turn caused an increase in surface and subsurface runoff [16,23]. Gullies were initiated in the Debre Mawi watershed at the locations that were springs 40 years ago [16]. Active gullies are found in the periodically saturated bottomland portion of the main watershed, and a two-meter deep gully emerged in the saturated portion of watershed W2 in 2017.

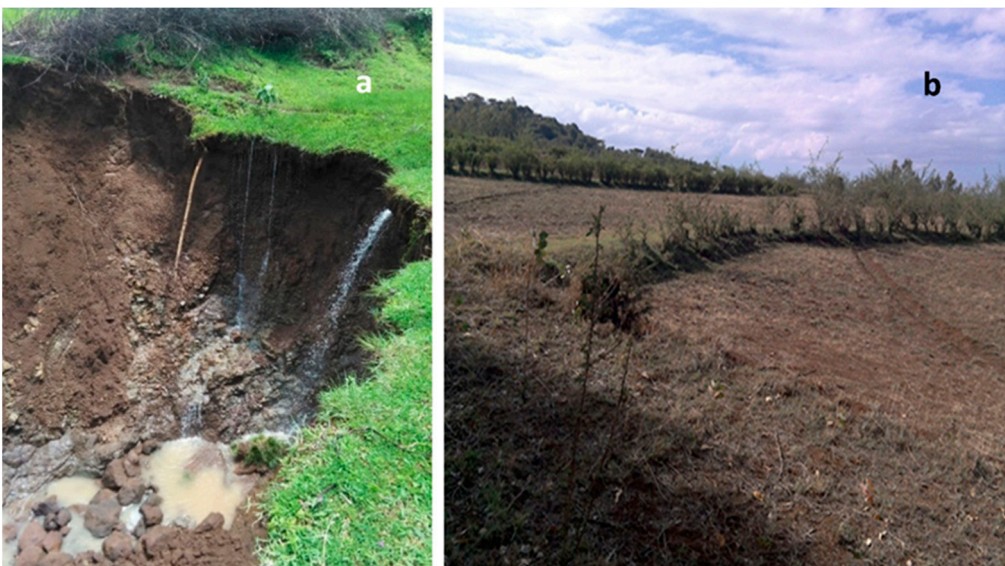

**Figure 3.** Photos depicting features in the Debre Mawi watershed: (**a**) Five-meter deep gully in the saturated bottomland of the Debre Mawi watershed. (**b**) Bund with *Sesbania grandiflora* in 2017. The bund, together with the infiltration furrow, was constructed in 2012. The infiltration furrow was filled with sediment in 2018 and is not visible.

Starting early in 2012 and ending in 2014, the government mandated SWCPs to be implemented by farmers as part of a national campaign. The SWCPs consisted of bunds with infiltration furrows that were installed on the contour on cultivated and grazing lands with slopes ranging from 3% to 32% according to guidelines of the Ethiopian Ministry of Agriculture. The infiltration furrows were 0.5 m deep and 0.5 m wide. The bunds varied in height from 0.3 to 0.6 m. The horizontal spacing between the bunds was 32 m. The spacing was reduced on steep lands so that the maximum difference in elevation was 1.5 m. Sesbania grandiflora was planted on the bunds as animal fodder and for the strengthening of the sides (Figure 3b).

As we observed from Google Images at the end of the implementation period in 2014, 70% of the watershed was treated with SWCPs. However, they were not maintained. Consequently, in the saturated bottomland areas within a three-year period, most furrows were filled with sediment, and only the bunds were visibly covered with grass (Figure 4a). In one case, where the rate of filling was less than the transport capacity of the sediment out of the furrow, the furrow acted as a cutoff drain. The concentrated flow resulted in a gully (Figure 4b) that formed in 2013 and then cut out a path downstream in the remainder of the years. The infiltration furrows on the sloping lands in the remainder of the watershed filled with sediment more slowly. By 2018, all were filled with sediment, and only the bunds remained visible as green strips (Figure 4a). In addition, without government support, farmers voluntarily installed off-contour traditional furrows (or fesses in Amharic) after the plowing was finished. These served to remove excess rainfall. The dimensions were determined by the size of the local ox-driven plows (*Marsha*) and were approximately 20–30 cm wide and 10–15 cm deep. Local farmers report that fesses were preferred because, unlike deep furrows, they do not hinder tillage operation.

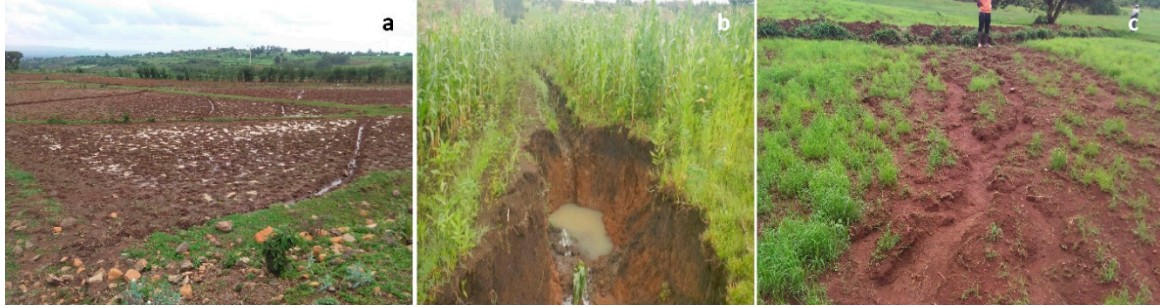

**Figure 4.** Soil and water conservation practices. (**a**) Fesses (plowed off-contour furrows; 20–30 cm wide and 10–15 cm deep) and 20–30 cm high bunds (green strips) in the Debre Mawi watershed in July 2018. Water is ponded at the surface, and the fesses carry off the excess water. Infiltration furrows uphill of the bunds have been filled with sediment and are not visible. (**b**) Infiltration furrow installed in 2012 in the saturated bottomland carries off interflow from the upland and caused the 2.5 m deep gully in the foreground. The photo was taken in August 2015. (**c**) Fifty to sixty centimeter wide rills formed by runoff concentrated by a bund on 18 July 2017.

Several other activities took place in the watershed, such as upgrading the unpaved road to a paved highway with a stone drainage ditch. The construction started in 2014 and lasted until 2016. The road crosses the watershed in the northern part of the eastern boundary (Figure 2). The unpaved road drainage ditch discharged into watershed W2. Some of the runoff came from the watershed across the road. The center of the paved road was distinctly higher than the unpaved road and prevented the water from crossing the road [15]. The exact amount of discharge is not known and varied during construction. Watershed W3 did not receive road drainage. Some drainage entered watershed W5 just north of watershed W3 (Figure 2). The weirs 1, 2, and 3 at the outlet of watersheds W1, W2, and W3, respectively, were located above a volcanic dike that interrupted the interflow, and the water table came to the surface. The outlet of watershed W4 is below watershed W3 on the same drainage path. The land between weir 3 and weir 4 is relatively flat and, therefore, subject to saturation during the rain phase [33]. Hence, all watersheds had saturated areas above the weir during the rain phase, which was covered by grass that tolerates water tables at shallow depths (Table 1). The acreage of grassland indicated the periodically saturated area in watershed W1 above weir 1 was the largest, and watershed W3 had the smallest. Watershed 2 received an unknown amount of storm runoff from the main road.

*2.2. Data Collection*

*Precipitation:* The five-minute rainfall amounts were recorded with an automatic tipping bucket rain gauge in the center of the watershed W5. The rain gauge was located at 37°25′21″ E and 11°21′31″ N. Precipitation was measured during the rain phase (June to October). Data from the Adet Agricultural Research Center 7 km south of the watershed were used to fill the 4 days missing in 2015 data and the 3 days in 2017.

*Stream Flow*: Runoff was measured during rain events from June to September for each of the five weirs. Plot scale measurements were not considered. Flow depth and velocity were measured at 10-min intervals from the time that the water became turbid until the water was clear. The surface velocity was determined by measuring the velocity of a float inserted 5 m upstream of the weirs, by recording the time taken by the float to reach the weir. The discharge was estimated as the product of a cross-sectional area and 2/3 of the surface velocity [34].

*Suspended Sediment Concentration:* One-liter water samples for sediment analysis were collected directly after the stage height measurement. The sediment concentration was determined by filtering the sample using filter papers with a pore size of 2.5 μm. The weight of the sediment was determined after drying for 24 h at 105 °C;

*Eucalyptus expansion:* Google Earth Image and field observation were used to map the expansion of the eucalyptus trees in the watershed. ArcGIS 10.3 was used for spatial data analysis.

### 2.3. Data Analysis

The procedure by [20] was used to analyze hysteresis in the sediment–discharge relationship for all storm events with five or more observations. First, discharge and sediment concentration are plotted over time. The loops can then be identified by comparing the timing of the sediment and discharge peaks. Mixed loops had two or more sediment peaks with at least one before and after the discharge peak. For clockwise loops, the only sediment peak occurred before the discharge peak, and for counter-clockwise loops, the sediment peak was after the discharge peak.

Other excel based statistical tools used were the Mann–Kendall trend test [35] to examine the trends in precipitation, sediment concentration, and runoff ratios, and the ANOVA (F test) for temporal and spatial changes in sediment concentrations between weirs.

## 3. Results and Discussion

### 3.1. Precipitation and Discharge

Precipitation during the rain phase (June-September) varied from a minimum of 832 mm in 2012 to a maximum of 1040 mm in 2017 (Figure 5). The maximum daily precipitation was 152 mm on 11 June 2014. The precipitation did not have a significant trend from 2010 to 2018, as confirmed by a Mann–Kendall [35] trend test.

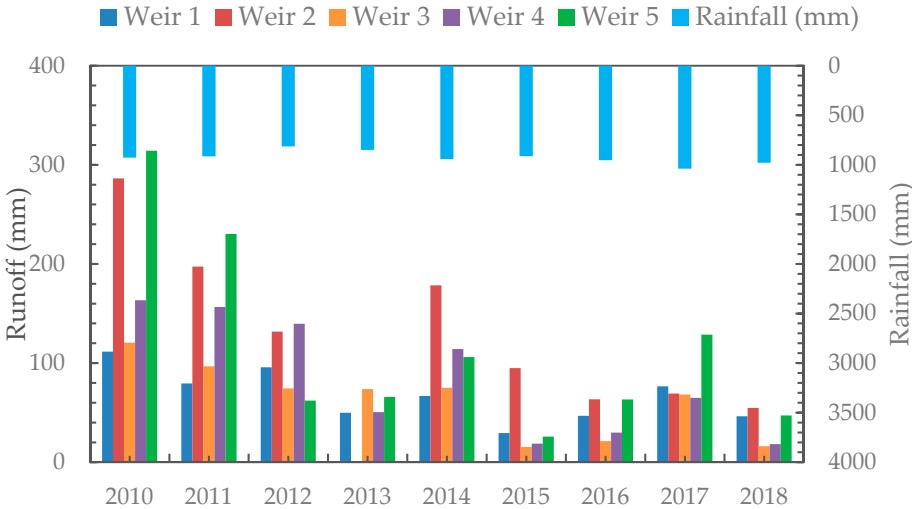

**Figure 5.** Precipitation and discharge in Debre Mawi watershed from 2010 to 2018.

The annual direct runoff for the main and nested watersheds was greatest in 2010 and 2011 before practices were installed, and then it decreased (Figure 5). The decrease in direct runoff was statistically significant (Mann–Kendall trend test at 5% level) [35]. In 2010, the maximum annual runoff depth was 314 mm during the rain phase for watershed W5 and 275 mm in watershed W2. The smallest annual direct runoff was in 2015.

The runoff coefficient is a good measure for assessing changes in hydrology. The ratio is defined as the discharge divided by the precipitation during the rain phase. Unlike discharge, it is minimally sensitive to precipitation amounts. Figure 6 shows that all runoff coefficients were decreasing from 2010 to 2015 and then remained the same or inclined slightly from 2016 through 2018. The minimum according to the exponential fit of the average runoff in all watersheds was in June 2016. The decrease in runoff means that more of the rainfall infiltrated and less became runoff. Evaporation is at the potential rate during the rainy season and will not affect the runoff coefficient [36].

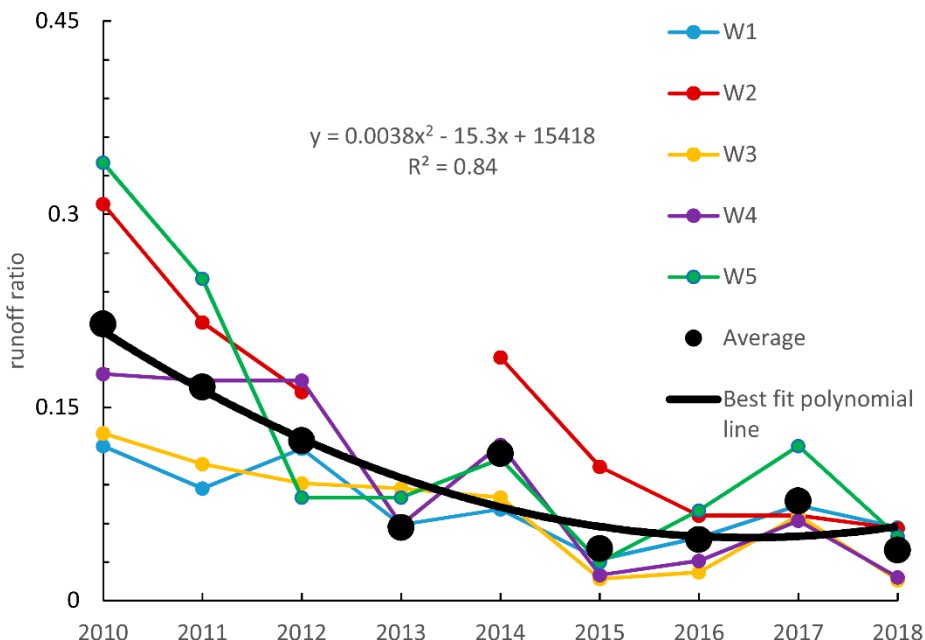

**Figure 6.** Runoff coefficients during the rain phase from 2010 to 2018 for the Debre Mawi watershed. Note watershed W2 was not monitored in 2013. The black line is the best polynomial fit line of the average runoff ratio of the five watersheds.

The runoff pattern of watersheds W2 and W5 was different than the other three nested watersheds W1, W3, and W4 (Figures 5 and 6). However, only before 2016 in watershed W2 was the discharge and runoff ratio significantly greater than the other nested watersheds because the drainage water from the unpaved road drained through watershed W2 [15]. After the road construction ended in 2016, the annual discharge of W2 was statistically the same as the other nested watersheds.

Both the runoff depth per unit area of watershed and the runoff ratio for the main watershed W5 was generally greater than the nested watersheds without the road drainage W1, W3, and W4 during high rainfall years (for example, 2010 and 2017; Figures 5 and 6). During moderate rainfall years (2015, for example), the discharge per unit area of W5 was nearly equal to the nested watersheds W1, W3, and W4. In 2012, the driest year, discharge per unit area of the main watershed W5 was less than any of the nested watersheds (Figure 5). This is likely due to the hydrological behavior of the grassed valley bottom area above weir 5 that becomes saturated during the rainy season. This area is a source of runoff during the wetter years [33], but we hypothesize that in a dry year, such as 2012, it acted as a sink for the runoff of the upper watershed rather than a source for runoff.

Another good measure to assess the changes in the watersheds is the amount of rain needed after the dry phase for the first runoff to occur. Figure 7 shows for all the watersheds the effective cumulative rain needed since the beginning of the rain phase to obtain 3 mm or more cumulative runoff. Effective rainfall is defined as rainfall minus potential evapotranspiration. Cumulative rainfall to generate runoff increased throughout the duration of the experiment, but the increase is not the same for all watersheds. The slope of the regression lines indicated that for watershed W1 and W2, the rainfall before runoff occurs increased by 30–35 mm $a^{-1}$ and for the other three watersheds between 50–60 mm $a^{-1}$ (Figure 7). Before the expansion of eucalyptus tree in watershed W3, the cumulative rainfall to generate 3 mm cumulative runoff (Figure 7) was less than W2 and W1, but after the great expansion of the trees in watershed W3 (Figure 2), it required higher amount of cumulative runoff to generate 3 mm cumulative runoff (Figure 7) than the two nested watersheds W1 and W2.

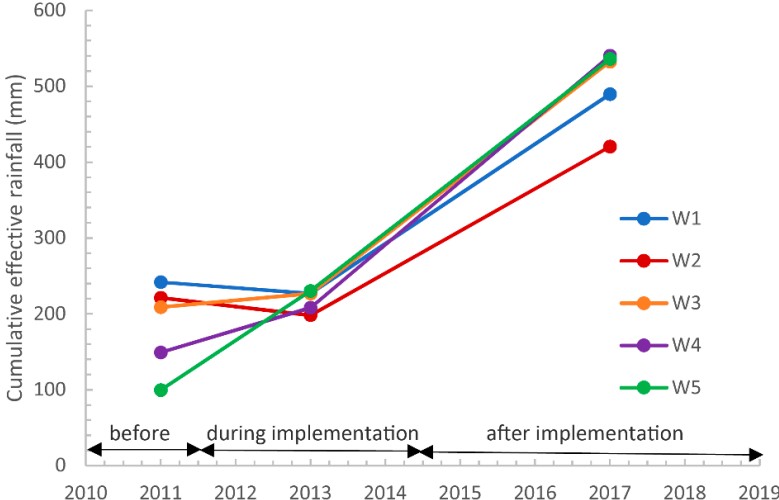

**Figure 7.** Average cumulative rainfall required to generate greater or equal to 3 mm runoff before (2010–2011), during (2012–2014), and after (2015–2018) SWCPs and eucalyptus tree expansion.

### 3.2. Effect of Changes in the Watershed on Discharge

To explain these changes in the runoff patterns, there are three likely causes, as discussed above, implementation of the SWCPs, increase in eucalyptus tree acreage, change in road drainage from a paved road, and precipitation. Since changes in runoff at the outlet are integral to all processes that take place within the watershed, it is difficult to sort out the exact cause with certainty. However, comparing the differential in runoff ratio and the amount of storage after the dry phase together with our observations, some tentative interpretations are possible. We will first discuss the SWCPs, followed by the eucalyptus trees and finally the road.

#### 3.2.1. Soil and Water Conservation Practices

The government imposed SWCPs were installed in the period from 2012–2014. However, maintenance was poor, and consequently, during the five-year period after SWCP implementation, all infiltration furrows filled up, and only the bunds covered with grass remained visible (Figure 4a). In the periodically saturated bottomlands, SWCPs initiated a gully (Figure 4b) and in the uplands flow was concentrated by the bunds and caused erosion in some locations (Figure 4c) The long-term decrease in runoff at the watershed scale, as shown in Figures 5 and 6, was not expected because of the poor maintenance of the SWCPs and especially based on findings in the 99 ha Anjeni watershed where the implementation of off-contour infiltration furrows had a minimal impact on runoff [37]. In Debre Mawi watershed, the infiltration furrows were on the contour, and the initial short-term decrease in runoff in 2015 could be explained. However, because the infiltration furrows started to be filled with sediment (Figure 4a), the capacity to store water in the infiltration furrows decreased. This should lead to a gradual increase in runoff to levels before the installation, such as observed in the Anjeni watershed. However, that was not the case in the Debre Mawi watershed: The runoff ratio and runoff stayed below the 2010/2011 levels. The reason is the increasing amount of water removed during the dry phase by evapotranspiration by the expanding acreage of the eucalyptus trees, as we will argue below.

#### 3.2.2. Eucalyptus Trees

Eucalyptus acreage increased three-fold over the nine years that the streamflow was measured. Before our observations started in 2010, 1.5 ha of trees were located on the most erodible lands in watershed W5 on the main stem below watershed W1 where crop production was not judged profitable by the farmers (Table 1, Figure 2). All were young seedlings in 2010 (refer to Figure 2). Plantings after

2010 were mostly on cropped lands, as the increasing need for charcoal and building material in the cities made eucalyptus cultivation more profitable than annual crops. In 2018, eucalyptus tree coverage was a total of 5 ha with 1.13 ha in watershed W3, 0.74 ha in watershed W2, 0.4 ha in watershed W4, and minimal in watershed W1 (Table 1, Figure 2).

Research in the Ethiopia highlands has shown that the increase in eucalyptus will increase water removal by evapotranspiration from the watershed during the dry phase [38–40]. A study in the Fogera Plain near Lake Tana found that the evapotranspiration of eucalyptus during the dry season was twice the potential evaporation of 4–5 mm/day due to additional energy of the dry wind [38]. Similarly, experiments south of Lake Tana found that during the dry phase, eucalyptus decreased the water content of the soil faster than native trees in a similar watershed south of Lake Tana [39]. Finally, in a watershed study in the central Ethiopian highlands, the shift from agricultural fields to eucalyptus reduced surface runoff by 21% [40].

The water that is removed from the watershed during the dry season needs to be filled up before runoff occurs. We proposed that the delay in runoff is caused by this phenomenon. The conceptual argument is as follows: The infiltration rates are high in the Debre Mawi watershed [41], and therefore, runoff occurs only when the soil becomes saturated [41,42]. Saturation of the soil occurs when the amount of rainfall during the rain phase equals the amount of water removed during the dry phase. Thus, a later start to runoff during the wet season indicates a larger amount of water removed during the dry season.

Thus, assuming that the dry season lasts for 200 days and the enhanced evapotranspiration of an area of eucalyptus trees is 10 mm d$^{-1}$, this area of eucalyptus trees can potentially evaporate 2 m a$^{-1}$ of soil water. Since 5% of the area in watershed W5 consists of the eucalyptus trees, the 2 m of evaporation of water over 5% areas average to 100 mm over 100% of the watershed. A similar calculation shows that in watershed W3, where the eucalyptus trees take up 20% of the area, the average evaporation by the eucalyptus accounts for approximately 400 mm over watershed W3. Thus, the decrease in storage noted in Figure 7 can be only partially ascribed to the eucalyptus trees except for W3, where the 50 mm a$^{-1}$ over nine years is approximately equal to the 450 mm. The decrease in runoff, as shown in Figure 5, is in the same order as the amount of water removed by the eucalyptus trees.

The runoff ratio of weir 4 and weir 3 decreased in time. (Figure 8). Weir 4 is located below weir 3 on the same stream. A flat area between the two weirs saturated in the rain monsoon phase. The explanation of the decreasing runoff ratio in Figure 8 is as follows: The discharge from watershed W3 decreased with the expansion of the eucalyptus trees. The reduced discharge decreased the extent of the saturated area in watershed W4. This, in turn, decreased the saturation excess runoff from watershed W4 through weir 4 further than the incoming recharge from watershed W3. Supporting this argument is that in 2013, a year with low rainfall and discharge, the runoff ratio of weir 4 and 3 was only 0.6 indicating that in that year the extent of the saturated area was the smallest when the runoff of watershed W3 was the smallest (Figure 8)

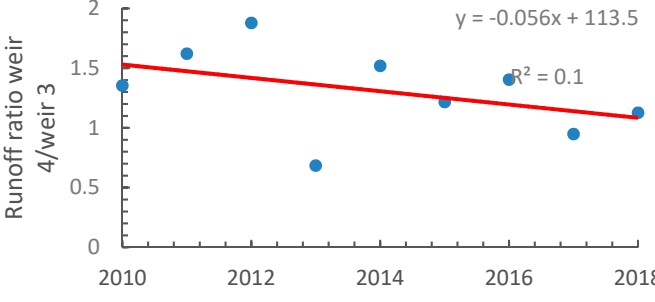

**Figure 8.** Runoff ratio of weir 4/weir 3 over time for the Debre Mawi watershed.

### 3.2.3. Road Construction

Finally, road construction changed the hydrology of watersheds W1 and W2 after 2014. The runoff coefficient of watershed W2 was much greater than the nested watersheds from 2010 to 2015 (Figures 5 and 6), and it was similar from 2016 to 2018. The runoff coefficient for watershed W1 decreased less than that of other watersheds over the nine years (Figures 5 and 6), indicating that another source of water became available in watershed W1. This change was likely caused by the drainage pattern of the paved road, which had a much higher centerline than the unpaved road. Runoff water that flowed over the unpaved road to watershed W2 was blocked by the higher paved road.

### 3.3. Sediment

#### 3.3.1. Sediment Load

The annual soil loss per unit area followed the same overall pattern as the direct runoff. The sediment loss was calculated by summing the product of discharge and sediment concentration and dividing by the area of the watershed. On average, 21 storms were recorded in each rainy year from all weirs. The soil losses were greatest during the first two years (Table 2). For example, the maximum yield for the entire watershed W5 in 2010 was 70 Mg ha$^{-1}$ and then decreased to values ranging from 0.2 to 16 Mg ha$^{-1}$. The sediment yield per unit area was much greater from the entire watershed than its parts. The difference was larger during wet years than in dry years. In almost all the study years, watershed W2 had greater soil losses than the W1, W3, and W4 due to the road drainage and the formation of a gully.

**Table 2.** Annual and average soil loss (Mg ha$^{-1}$ a$^{-1}$) and standard deviation (st dev) for the Debre Mawi watershed. Soil and water conservation practices (SWCPs) were installed from 2012 to 2014.

| Watershed | 2010 | 2011 | 2012 | 2013 | 2014 | 2015 | 2016 | 2017 | 2018 | Average | St dev |
|---|---|---|---|---|---|---|---|---|---|---|---|
| W1 | 3.1 | 3.4 | 2.6 | 1.2 | 1.7 | 0.4 | 1.2 | 1.9 | 0.2 | 1.7 | 1.1 |
| W2 | 18.5 | 13.7 | 4.3 | - | 8.2 | 1.1 | 2.2 | 3.4 | 0.3 | 6.5 | 6.5 |
| W3 | 5.2 | 8 | 2.4 | 2.6 | 2.7 | 0.2 | 0.5 | 2.9 | 0.2 | 2.7 | 2.5 |
| W4 | 12 | 19.9 | 5.1 | 1.8 | 4.7 | 0.4 | 0.8 | 4.1 | 1.4 | 5.6 | 6.4 |
| W5 | 70.3 | 53.9 | 9.0 | 13.3 | 12.5 | 0.3 | 4.1 | 15.8 | 2.8 | 20.2 | 24.6 |
| Average | 21.8 | 19.8 | 4.7 | 4.7 | 6.0 | 0.5 | 1.8 | 5.6 | 1.0 | | |
| St dev | 27.8 | 20.1 | 2.7 | 5.7 | 4.4 | 0.4 | 1.5 | 5.7 | 1.1 | | |

#### 3.3.2. Sediment Concentration

The average annual suspended sediment concentration (Table 3) decreased over the nine years to concentrations that were approximately half of the 2010 concentrations. Sediment concentrations were the smallest in 2015, which was a dry year at the end of the period of implementing SWCPs. Concentrations at the outlet of the entire watershed were greater than its upland parts, indicating that sediment was picked up in the valley bottom. Except for watershed W2 in 2011, watershed W4 (which had an active gully in the first years) had a greater concentration than the nested watersheds W1, W2, and W3 during these years. The W4 gully became stable over the course of the experimental period. In the next sections, we will further analyze the reasons for the change in sediment concentrations over the nine years.

**Table 3.** Suspended sediment concentration (g L$^{-1}$) for the main watershed W5 and the nested watersheds (W1–W4).

| Watershed | 2010 | 2011 | 2012 | 2013 | 2014 | 2015 | 2016 | 2017 | 2018 |
|---|---|---|---|---|---|---|---|---|---|
| W1 | 3.3 | 3.7 | 2.4 | 2.3 | 2.8 | 1.5 | 2.9 | 1.6 | 0.4 |
| W2 | 5.7 | 6.3 | 3.2 | - | 3.6 | 1.3 | 3.0 | 3.0 | 1.4 |
| W3 | 3.5 | 4.3 | 3.1 | 3.3 | 3.8 | 1.0 | 2.8 | 2.2 | 1.2 |
| W4 | 6.2 | 5.5 | 3.7 | 3.3 | 3.6 | 2.0 | 2.8 | 2.0 | 0.6 |
| W5 | 12.7 | 13.0 | 13.1 | 11.5 | 8.2 | 4.2 | 5.8 | 7.7 | 5.7 |

### 3.4. Discharge–Sediment Concentration Hysteresis Patterns

For the five watersheds, a total of 590 runoff events with a minimum of five observations at ten-minute intervals of discharge and suspended sediment concentration pairs were analyzed for determining the relationship between the suspended sediment concentration and discharge. By plotting the sediment discharge pairs, three types of loops were identified: clockwise, counter-clockwise, and mixed. Examples are given in Figure 9. In general, clockwise and counter-clockwise loops are characterized by systematic offsets in the time of peak discharge and the time of peak sediment concentrations. In mixed loops, the concentrations during the storm did not vary greatly.

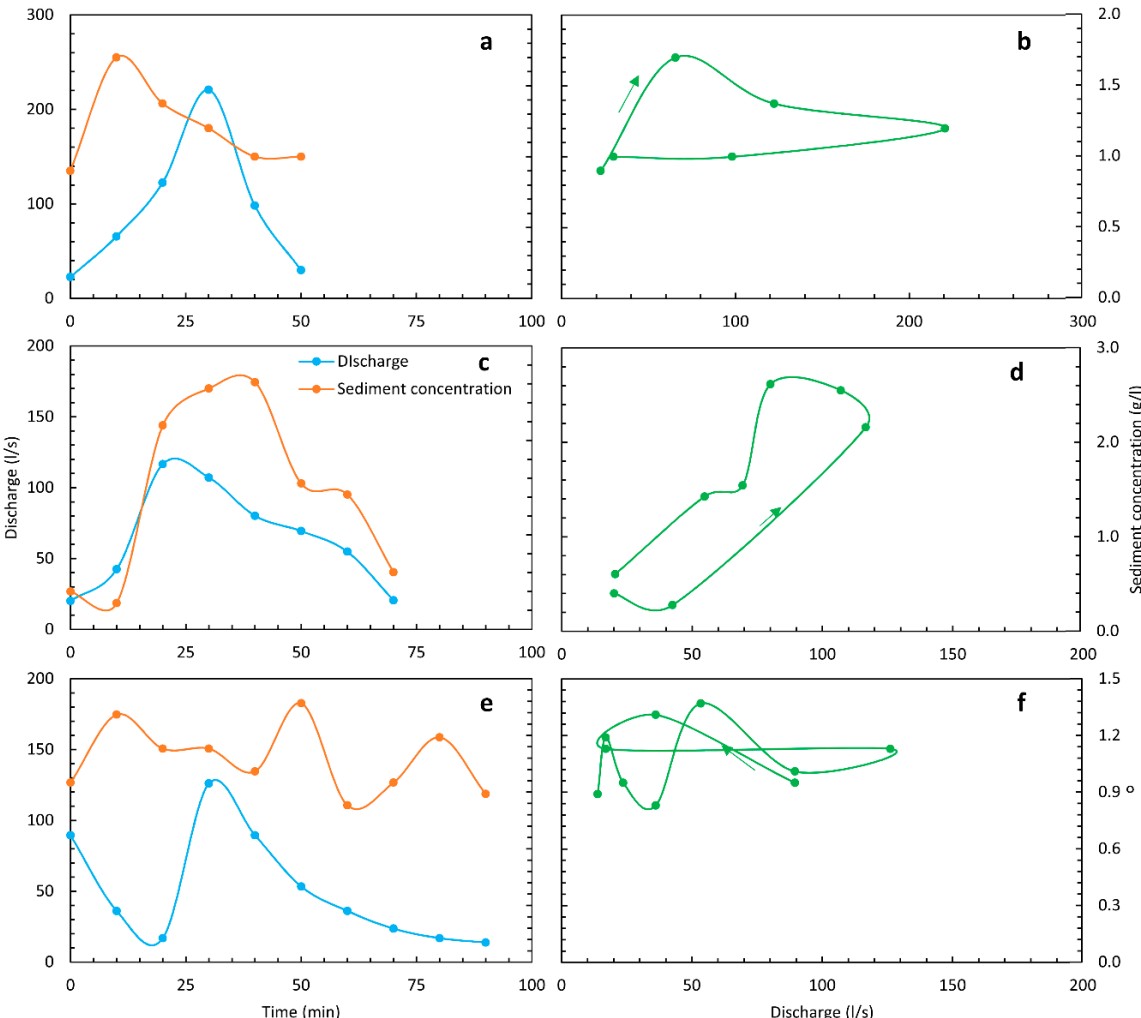

**Figure 9.** Representative examples of hydrographs and sedigraphs (**left**) and hysteresis loops (**right**) for the Debre Mawi watershed during the rainy season: (**a**,**b**), clockwise pattern at the outlet of the main watershed W5 on 27 August 2017; (**c**,**d**), counter-clockwise pattern at the outlet of the nested watershed W1 on 1 August 2012; and (**e**,**f**), mixed loop pattern, watershed W2 on 6 July 2016.

The percentage occurrence of each typical loop for all the years in each watershed is shown in Figure 10, and the total number of loops for each year for all five watersheds is in Figure S1 in the Supplementary Material. The number of loops decreased over the years, because the discharge and the duration of the storms decreased so that there were fewer storms with five or more paired measurements of discharge and sediment concentrations. Of the storms analyzed, the mixed loops with no clear pattern between runoff and sediment concentration (Figure 10c) were most common for the early years and became approximately equal to the number of clockwise loops in 2015 and

later (compare Figure 10a,c). The number of counter-clockwise loops decreased over the study period (Figure 10b).

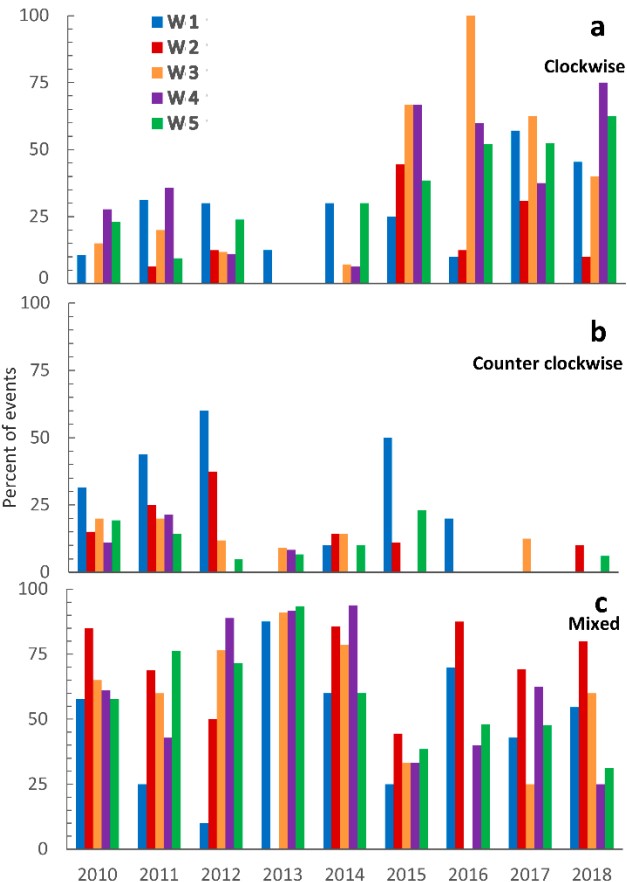

**Figure 10.** Hysteresis in the sediment concentration and discharge at the outlet of the four nested watersheds W1–W4) and the main watershed W5 of Debre Mawi watershed. (**a**) Clockwise loops; (**b**) Counter-clockwise loops, and (**c**) Mixed loops. The location of the watersheds is given in Figure 1.

*3.5. Effect of Changes in the Watershed on Sediment Transport*

3.5.1. Effect of SWCPs and Gullies on Sediment Transport

SWCPs: After the implementation of SWCPs, sediment concentrations and load significantly decreased in all the five watersheds. The decrease in sediment load, especially during the SWCPs implementation years (2012–2014), was caused by the decrease in runoff that has been trapped by the infiltration furrows. This agrees with [10] and [37] that indicated SWCPs were effective in the short-term in humid watersheds. However, in the long term, infiltration furrows were not effective in reducing soil loss because most infiltration furrows were filled with sediment, as shown in Figure 4a. The bunds concentrated the flow and caused rill formation downstream of the bunds (Figure 4c).

Gullies can affect sediment load and concentration. The sediment concentrations in the main watershed W5 were generally greater than the other nested watersheds because of the pickup of sediment in the gullies [16]. The 1500 $m^2$ gully, especially in watershed W5, was contributing to the large portion of catchment soil loss (Table 1). A recent study in the Debre Mawi watershed found that that 92% of the sediment at the outlet of a gully originated from the gully itself [23].

### 3.5.2. Effect SWCPs and Gullies on Discharge–Sediment Concentration Relationship

To understand the effect of SWCPs on discharge–sediment concentration relationship, we divided the storm event sediment loops during the nine-year period (shown in Figure 10) into three blocks: Period I (2010 and 2011) before SWCPs implementation; Period II, consisting of the years 2012, 2013, and 2014 during implementation of SWCPs; and Period III, covering the last four years 2015, 2016, 2017, and 2018 after SWCPs implementation and during road construction (Table 4). In Period I, before 2012, out of the 185 recorded storm events with five or more sediment concentration observations for all five watersheds, 17% were clockwise, 22% counter-clockwise, and 61% showed mixed patterns (Table 4). In Period II, of the 212 storm events, the mixed loops increased to 72% with approximately the same number of clockwise and counter-clockwise loops. After the implementation of SWCPs, of 193 events, the mixed loops reduced to 48%, clockwise 45%, and 7% counter-clockwise patterns.

**Table 4.** Percent of loop types before, during, and after installation of SWCPs.

| Type of Loops | 2010–2011 | 2012–2014 | 2015–2018 |
|---|---|---|---|
| Clockwise | 17 | 14 | 45 |
| Counter-clockwise | 22 | 15 | 7 |
| Mixed | 61 | 72 | 48 |

The large number of mixed hysteresis loops (Table 4) during the implementation of the soil and water conservation practices (Period II) indicates that unconsolidated soil from the newly constructed bunds was available throughout the rainstorm that could be transported by runoff. Thus, the sediment concentration was elevated during both the rising and falling limbs of the hydrograph, as shown in Figure 9e. In addition, the mixed loops were associated with the largest runoff events. It is not incidental that the longest-lasting storm in Figure 9 had a mixed loop.

The increase in the clockwise loops and the decrease in counter-clockwise loops from Period I to Period III in Table 4 indicates that a change in timing of when transport occurs from later in storm to earlier in storm that is partly due to less sediment available for transport as indicated by the temporally decreasing sediment concentrations (Table 3). The increase in clockwise can also be explained by the stored sediment in the channel from the previous rainstorm that becomes the source of sediment on the rising limb of the new event [24,43,44].

The counter-clockwise hysteresis pattern occurs when the sediment source is at a distance from the measurement location. Watershed W1 had the greatest number of counter-clockwise loops in Period I and the beginning of Period II (Figure 10b; Table 4). This watershed has a large grassy area in front of the weir, and agricultural land begins at 175 m above the weir. We hypothesize that the first runoff to reach the weir during the rainstorm is generated from the grassy area, which is perpetually saturated during the rainy season. This runoff water has a low sediment concentration because of the grass. The saturated area expands during the rainstorm and will expand in the cropland during a large enough storm, increasing the sediment concentration in the water. If the intensity of the rain decreases towards the end of a storm, the runoff will decrease, but the sediment concentration still will be high, generating a peak in the sediment concentrations after the peak of the runoff. The other watersheds do not show this pattern because their cropped area is much larger than any grassy area. Thus, the sediment signal of the grassy area is hidden by the rill erosion of cropped land.

Bottomland gullies could also affect discharge–sediment patterns. For instance, the sediment available through gully bank failure by the previous storm can result in a clockwise loop in that the sediment peak reaches the outlet before the discharge peak [23].

In addition to the SWCPs and gullies, another likely reason for our finding that soil loss was reduced over the course of the study period might be the hysteresis analysis method itself. For example, if we take a closer look at Figure 9e, it is obvious that the sediment supply becomes limiting at the end of the rainstorm because the flow increased while the sediment concentration remained low. This is

unlike at the beginning of the storm. According to our classification scheme, this should be a clockwise loop, while the result of the analysis is a mixed loop (Figure 9f).

## 4. Conclusions

This study investigated, over nine years, the impact of soil and water conservation practices and the expansion of acreage of eucalyptus trees on the hydrology and soil loss in an agricultural watershed in the 95 ha Debre Mawi watershed in the Ethiopian highlands. The watershed is of volcanic origin, and lava dikes block the flow of subsurface water at several locations forcing the subsurface flow to the surface and causing periodically saturated areas. The saturated areas are the source of the surface runoff. Soil and water conservation practices consisted of 50 cm deep infiltration furrows with bunds downhill. Maintenance was poor, and infiltration furrows were filled up at the end of the study. The results show that the direct runoff and sediment decreased by a factor of two or three over the nine years. Although further studies are needed, it seemed that the evapotranspiration of eucalyptus during the dry season was mainly responsible for the reduction in the direct runoff by creating storage for the rainwater to infiltrate. This is in accordance with local knowledge that wetlands dry up after eucalyptus were planted. Soil loss reductions were mainly related to the smaller amounts of runoff.

The implication of the research is that soil and water conservation practices are effective in the short term but likely not as effective in the long-term after they are filled up with sediment. Expansion of eucalyptus trees in a watershed reduces direct runoff and erosion from saturated areas in the watersheds of the sub-humid Ethiopian highlands.

**Supplementary Materials:** The following are available online at http://www.mdpi.com/2073-4441/11/11/2299/s1. Figure S1. Number of loop types in time in the Debre Mawi Watershed over a nine-year period from 2010–2019.

**Author Contributions:** D.A.M., D.C.D., T.C.A., C.D.G., and S.A.T.; collected; analyzed; wrote the manuscript. D.C.D., S.A.T., B.F.Z., and T.S.S.; analyzed; supervised and advised all the research work that led to this paper. T.S.S. and B.F.Z. revised and conducted language editing.

**Funding:** Funding for this research was provided in part by the Belmont Forum NILE-NEXUS project through the United States NSF award ICER-1624335. Additional funding was also obtained from the Blue Nile Water Institute, Bahir Dar University.

**Acknowledgments:** We are grateful for the data made available to this research from the PEER Science project of Bahir Dar University.

**Conflicts of Interest:** The authors declare that they do not have a conflict of interest.

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
