# Peer review of "Impact of Soil Conservation and Eucalyptus on Hydrology and Soil Loss in the Ethiopian Highlands"

_water, doi:10.3390/w11112299_

Round 1
Reviewer 1 Report
Title: Impact of Soil Conservation and Eucalyptus Trees on Hydrology and Soil Loss in the Ethiopian Highlands
This paper examines effects of three landscape changes (conversion to agro-forestry, sediment control measures, and road building) on discharge volume and sediment concentration in five watersheds in Ethiopia.
General remarks:
My main concern is about whether and how interactions between study watersheds are handled – watershed 1 is the largest, and watersheds 2-5 are contained within it. Consequently, patterns of erosion and discharge in W1 are dependent on those in W2-W5. Moreover, W3 appears to discharge into W4, and the impacts of runoff and sediment from W3 are not specifically discussed in relation to the temporal changes in runoff and sediment in W4. I’m not convinced that it is appropriate to compare the five watersheds because they are not distinct. At the very least, interactions between watersheds should be addressed in the paper’s discussion section.
Also, the impact of the road on discharge is unclear – what was the configuration of the ditches pre and post roadwork? Was the discharge changed (i.e., culverts installed to carry runoff from one side of the road to the other, potentially increasing inflow into the watershed)?
Finally, I am interested in the implications of the research. It is hard to tease out impacts of road construction, agro-forestry, and sediment control measures (especially the latter two) as they occur simultaneously. What is the main take-away message? Is agro-forestry helpful in reducing erosion and stream flow in this climate? Is this desirable given that dry season flows are also reduced? Discussion is somewhat vague on this. How can the results of the study be applied in practice or in contribution to the erosion knowledge base? It would be helpful to explicitly state these contributions.
Comments on figures and tables:
Figure 1: Please add units to the DEM. What are coordinates on image b? not geographic – UTM? Also, watershed 4 in this figure doesn’t appear to be aligned with the flow lines and boundary for W4 doesn’t appear to cover the full watershed area.
Figure 3: please add a scale to fig a, or add gully depth in the caption
Figure 4: Suggest move precipitation to a secondary y-axis so that differences in runoff depth can be discerned between watersheds. Why no data for W2 in 2013 (also in Fig 5)?
Figure 7: please add some indication of the scale in caption or on figures
Table 1: numbers are confusing because values in Eucalyptus column are variable, but those in the total column are not. Also, what are units? This table needs revision and explanation in the text.
Table 2: how were these numbers calculated? They appear to be averages of storms occurring during the year, but how many storms? What is average and variance of discharge and sediment yield for the storms by year and watershed? Descriptive statistics would be helpful along with appropriate discussion.
Table 3 and associated text: For most watersheds, sediment concentrations in 2018 were much less than half of the 2010 concentrations. Lowest concentrations occurred in 2015 and in 2018 – depends on watershed. Watershed 4 did not have the greater sediment concentration than others in all early years. In fact, it was only greater in 2010 and 2012, and tied with W3 in 2013. Clarification is needed.
Line by line remarks:
Line 39 add comma after overgrazing (use oxford comma throughout unless this conflicts with the journal’s style guide)
Line 47 remove “however”
Line 48 remove “first of”
Line 50 remove comma after Juu, add comma after bunds)
Line 54 “as a consequence of forest replacement” also rest of paragraph has inconsistent use of tense.
Line 87 geographic coordinates for outlet need seconds
Line 95 intrusive lava dykes – intrusive dikes are by definition igneous, so I don’t think the word lava is needed.
Line 129 remove “to”
Line 130 don’t begin the sentence with [15]. Try “In the Debre….. at locations that were springs 40 years ago [15].”
Line 153 add “cm” after 20-30
Line 156 configuration of ditch(es) is confusing. Here two ditches are indicated and on line 158 only one is indicated. Also “the exact amount” refers to what? To discharge or sediment production?
Line 166 consistent style needed “saturated area above Weir 1 was the largest and that above Weir 3 was the smallest.” Why say Watershed in one and Weir in the other?
Line 172 which watershed is refered to here? How much data were missing? Please provide number of days or percent of dataset that was contributed from the Research Center.
Line 194 to what does [35] refer?
Line 199 only five years of data represented here (2012 to 2017)? Why not full study time period?
Line 223 is confusing. Discharge in W5 was nearly equal to which watersheds? Sum of W1, W3, and W4? Or discharge per unit area? How can W5 be compared to W1-W4 given that they are not distinct.
Line 245 which are the two nested watersheds? W3 is also nested? This needs clarification.
Line 255 “our observations of the physical observations” is awkward. Try the word parameters, or omit everything after the first “observations”.
Line 265 remove word “in”
Line 278 Sentence beginning at end of line with “the capacity…” is long and difficult to follow. Consider breaking into two sentences.
Line 281 Why was this not the case? Can you propose an explanation?
Line 290 Relevance of eucalyptus forest east of watershed? I suggest this be omitted or moved elsewhere.
Lines 292-297: Avoid the use of reference #s at start of sentence. Try “Evapotranspiration of eucalyptus trees… water is available [38].” Also, how can ET be twice potential ET? Potential should be the maximum, actual ET may be up to or including Potential ET, but should not be greater. Throughout paragraph, please relate and compare the research, instead of reporting results from 3 papers sequentially. How are they relevant?
Lines 305- math is unclear. 200 days and 10 mm/day gives 2000 mm total, but is this per tree? How is the estimate upscaled to watershed? How is 2000mm (2 m) converted to 100 mm in line 307? Using the 5% number? If so, then 10mm/day is probably not a per tree statistic.
Line 311 equal, not equally
Line 315 paragraph – please use a consistent tense. Last sentence needs additional detail – what was the road pattern, how was drainage pattern changed? How does this influence runoff?
Line 349 sentence is unfinished
Line 372 suggest replacing “confirms the report by” with “agrees with”
Line 377 and paragraph – needs revision, structure is awkward and wording is repetitive.
Line 397 45 is used twice? Typo?
Line 403 “As shown in Figure 8” – I don’t see how the data presented in Fig 8 shows mixed loops were associated with larger runoff events.
Line 407 I don’t agree that Table 4 indicates sediment concentration decreased over period of observation. I think these results show a change in timing of when transport occurs (from later in storm to earlier in storm by the shift to clockwise hysteresis)
Line 439 replace “sediment” with “sediment transport”
Author Response
Thank you for the review. Our response is in the attached pdf
Thanks again
Demis and Tammo

Reviewer 2 Report
Overall this is an interesting study, with monitoring conducted over a series of years. The article is very well written and does a great job of conveying the research effort and results. I have minor comments to improve the overall quality of the paper:
in the abstract, "9" should be "nine".
Keywords: "Practices" should be lower cased
line 78: soil and water conservation practices were already defined as SWCPs, stay consistent and use the notation throughout. Check the entire paper for these instances.
Figure 2: I suggest you use different colors or hatching. Blue is typically used for water bodies.
line 153: units are missing
line 160: Weirs should be lower cased
line 349: the sentence cuts off, perhaps due to the figure. This needs to be fixed.
Table 4: capitalize "counterclockwise"
A better description, earlier in the paper, of the soil and water conservation practices used in the watershed will help readers make the connection on what improvements were made. In my opinion, Figure 7 in page 9, was too late to describe the practices. I would move this towards the front of the paper - in the introduction.
There are several instances where in-line citations replace descriptions or wording, for example: [38] found...., ...a study by [40] ... I don't believe this is an accepted method of writing with citations. The sentences should be complete and be readable without the in-line citation. The citation should be used to provide evidence of the statement being made.
Both the abstract and the conclusion paragraphs should be strengthened to include a summary of quantifiable data that was discussed in section 3.
Author Response
Dear Reviewer
Thank you for your comments. The response and the marked up manuscript are attached
Regards
Demis and Tammo
